# The Roles of Coenzyme Q in Disease: Direct and Indirect Involvement in Cellular Functions

**DOI:** 10.3390/ijms23010128

**Published:** 2021-12-23

**Authors:** Francesco Pallotti, Christian Bergamini, Costanza Lamperti, Romana Fato

**Affiliations:** 1Dipartimento di Medicina e Chirurgia, Università Degli Studi dell’Insubria, 21100 Varese, Italy; 2SSD Laboratorio Analisi-SMEL Specializzato in Citogenetica e Genetica Medica, ASST Settelaghi-Ospedale di Circolo-Fondazione Macchi, 21100 Varese, Italy; 3Dipartimento di Farmacia e Biotecnologie, FABIT, Università Degli Studi di Bologna, 40126 Bologna, Italy; romana.fato@unibo.it; 4UO Genetica Medica e Neurogenetica Fondazione IRCCS Istituto Neurologico C. Besta, 20133 Milano, Italy; Costanza.Lamperti@istituto-besta.it

**Keywords:** coenzyme Q10, ubiquinone-10, ubiquinol-10, mitochondria, OxPhos, LDL, statins, age-related diseases

## Abstract

Coenzyme Q (CoQ) is a key component of the respiratory chain of all eukaryotic cells. Its function is closely related to mitochondrial respiration, where it acts as an electron transporter. However, the cellular functions of coenzyme Q are multiple: it is present in all cell membranes, limiting the toxic effect of free radicals, it is a component of LDL, it is involved in the aging process, and its deficiency is linked to several diseases. Recently, it has been proposed that coenzyme Q contributes to suppressing ferroptosis, a type of iron-dependent programmed cell death characterized by lipid peroxidation. In this review, we report the latest hypotheses and theories analyzing the multiple functions of coenzyme Q. The complete knowledge of the various cellular CoQ functions is essential to provide a rational basis for its possible therapeutic use, not only in diseases characterized by primary CoQ deficiency, but also in large number of diseases in which its secondary deficiency has been found.

## 1. Introduction

Quinones are compounds widely present in nature, but their importance has increased considerably since they were recognized as fundamental components of the energy conservation systems present in all living organisms. The first quinone was discovered by Kofler, who was searching for compounds with Vitamin K activity in 1946 [1]. However, an accurate study of these molecules began only a decade later when ubiquinone(UQ), or Coenzyme Q (CoQ), was isolated in the laboratories of Richard Morton in Liverpool and David Green in Wisconsin. The presence of quinones had already been noted in biochemical studies on different phyla of living organisms. Crane and Lester’s studies on CoQ in electron transport led these authors to rediscover the role of Kofler’s quinone and to associate it with the electron transport mechanism in photosynthetic systems [2,3,4,5,6,7]

The chemiosmotic theory proposed by Mitchell in 1961 [8] introduced the revolutionary mechanism of transducing the energy associated with the protonmotive force into ATP synthesis, reinforcing the role of the CoQ in the mitochondrial electron transport chain. Indeed, the formation of trans-membrane ion gradients is the most ancient way to store energy in biological systems [9]. In fact, the presence of ATPases connected to redox enzymes was evidenced in LUCA (Last Universal Common Ancestor) and pre-LUCA organisms [10]. 

From the chemical point of view, ubiquinones show two important features: a redox active head (the benzoquinone ring) and a polyprenoid side chain varying in length between species. The side chain comprises ten isoprenoid subunits in humans (Ubiquinone-10), nine in mice (Ubiquinone-9), where ubiquinone-10 is also present as minor homolog, eight in *E. coli*, and six in *S. cerevisiae*. Although the electron and proton transfer activity is strictly associated with the benzoquinone ring, the polyprenoid tail is essential to restrict the molecule inside the lipophilic core of the biological membranes.

In addition to its bioenergetic role, CoQ has been associated with several important functions, such as the regulation of the cytosolic NAD+/NADH ratio and ascorbate reduction through the activity of the CoenzymeQ10 (CoQ10) dependent NADH-oxidase of the plasma membrane or PMOR, as it was originally termed by the researcher who first described it [11,12,13]. Then, after several years of research, it was discovered that Plasma Membrane Oxidase is a complex system, now indicated as Plasma Membrane Redox System (PMRS), involving different components; it serves as terminal oxidase for the cytosolic NADH and is responsible for the protein disulfide-thiol interchange activity on the outside of the plasma-membrane. Moreover, CoQ is a modulator of the mitochondrial transition permeability pore (mPTP) [14], is involved in gene regulation [15], and displays anti-inflammatory effects by influencing the expression of NFκ-β1-dependent genes [16,17,18,19]. In the reduced form, CoQ is a powerful antioxidant whose efficiency depends on its intramembranous distribution and the presence of many intracellular CoQ reducing enzymes. 

Finally, the presence of CoQ is essential to maintain the integrity and functions of cell membranes [20,21].

## 2. Ubiquinone/Coenzyme Q Biosynthesis

Ubiquinone/Coenzyme Q is the only vitamin-like compound that can be completely endogenously synthetized. The biosynthetic process of CoQ in mammals is a very complicated pathway that consists of three distinct steps, located respectively in the cytosol for the isoprenoid side chain and for the synthesis of the aromatic precursor of the benzoquinone ring and in the mitochondrial matrix for CoQ assembly and benzoquinone ring modifications. 

The isoprenoid chain is synthesized through the mevalonate pathway, which leads to the farnesyl pyrophosphate (FPP), a common precursor of various metabolites: cholesterol, dolichols, isoprenylated proteins, and Coenzyme Q [22]. The regulatory step of the mevalonate pathway involves the 3-hydroxy-3-methyl-glutaryl-CoA reductase (HMG-CoA), the molecular target for statins. Noticeably, a well-known side-effect of statin therapy is statin-induced myopathy (SIM), associated with mitochondrial dysfunction and related to a reduced level of CoQ10, both in plasma and in muscle tissues [23,24,25]. In eukaryotes, the terminal steps in the biosynthesis of CoQ are thought to be rate limiting and occur in the mitochondrial matrix. In yeast, the mitochondrial assembly of CoQ requires at least 11 proteins encoded by COQ genes organized in a biosynthetic complex [26]. The human homologues of some COQ yeast genes were identified and, currently, mutations in *COQ2* [27,28], *PDSS1* [29], *PDSS2* [30], *COQ4* [31], *COQ6* [32], *ADCK3* [33,34], and *COQ9* [35] have been described in patients with primary CoQ10 deficiency (OMIM # 607426).

The assembly step in CoQ biosynthesis is catalyzed by the enzyme COQ2, a polyprenyl transferase that binds the polyprenoid chain to the six position of the p-hydroxybenzoic acid. Once assembled, the aromatic head of this CoQ precursor is modified to lead to the mature CoQ molecule. All the enzymes involved in these modifications are organized in a complex indicated as CoQ-synthome, which is localized on the matrix face of the inner mitochondrial membrane; this localization raises the problem of the access of the hydrophobic precursor to the modifying enzymes. 

The term ubiquinone refers to the ubiquitous presence of this molecule in all biological membranes and, besides the presence of a homolog of COQ2 in the endoplasmic reticulum (UBIAD1) [36], there is no evidence for an extramitochondrial CoQ biosynthesis. Given the extremely high hydrophobicity of this molecule, the problem of its distribution remains unsolved. Recently, in the group of Pagliarini, two proteins were identified, named Cqd1 and Cqd2, located in the inner mitochondrial membrane and involved in ubiquinone trafficking. Loss of Cqd1 favors the extramitochondrial distribution of CoQ leading to a CoQ deficient syndrome, even if the total amount of CoQ remains unchanged [37].

### 2.1. Ubiquinone in Mitochondrial Electron Transfer Chains

Since its discovery, Ubiquinone or Coenzyme Q has been recognized as an obligate component of the mitochondrial respiratory chain. It is mainly localized in the inner mitochondrial membrane where it shuttles electrons from NADH-(Complex I) and Succinate-(Complex II) dehydrogenase to the bc1 complex (Complex III), then cytochrome c shuttles electrons to the cytochrome oxidase (Complex IV) directly responsible for O_2_ reduction. The free energy delivered by the electron transfer from the reduced coenzymes to the molecular oxygen drives the translocation of protons from the matrix site to the intermembrane space, generating the protonmotive force used by the mitochondrial ATPase to synthesize ATP. 

As a component of the electron transfer chains, ubiquinone can exist in three different redox states. The fully oxidized form, ubiquinone (CoQ), can be reduced by the two-step gain of two electrons and two protons to give the fully reduced form (ubiquinol form or CoQH_2_), passing through a radical form (the semiquinone form or CoQH) (Figure 1).

While the bioenergetic role of CoQ is clear, the structural organization of the respiratory chain and the mechanism by which electrons are transferred from reduced coenzymes to molecular oxygen are still a matter of debate. Following the observation that the concentration of ubiquinone was more than the concentration of the prosthetic groups in the redox enzymes, Green and Tzagoloff first introduced the idea of CoQ as a mobile electron carrier [38]. This hypothesis was supported by the pioneering work of Kroeger and Klingenberg [39], who derived a simple equation to describe the kinetic behavior of ubiquinone, concluding that all the quinones present in the inner mitochondrial membrane behave like a common pool and the rate of electron transfer depends only on the rate of input and output of electrons to the quinone’s pool. For the Random Diffusion Model of Electron Transfer chain [40], the redox enzymes are randomly distributed into the lipid bilayer and electron transfer is ensured by free diffusion of ubiquinone and cytochrome c. Nevertheless, about 15 years later, Schagger and Pfeiffer reintroduced the hypothesis of a solid-state organization of the respiratory chain based on structural evidence by blue native electrophoresis [41,42]. Subsequently, several pieces of evidence favoring alternatively solid-state organization or random distribution appeared in literature, leaving the problem unsolved. 

Complex II is not recognized as a component of this supramolecular organization. The original hypothesis of the common quinone pool, introduced by Kroeger and Klingenberg [39] in 1971, was replaced by assuming the presence of two different quinone pools: one acting for the oxidation of NADH and the second for the oxidation of succinate and for the oxidation of substrates of other dehydrogenases that use ubiquinone as an electron acceptor [43,44]. However, no quinones were found permanently bound to respiratory enzymes, leading to speculation of a dynamic equilibrium between quinone pools.

Although the role of UQ is to transfer electrons from dehydrogenases to the bc1 Complex, in the presence of a high protonmotive force and a high level of reduced ubiquinone (CoQH_2_) it is possible to observe a back-flow of electrons from ubiquinol to NAD+ involving Complex I. This Reverse Electron Transfer (RET) has been recognized since the second half of the 1960s [45], but it was considered an in vitro phenomenon observed in isolated mitochondria without physiological relevance [46,47,48]. RET was associated with an increased Reactive Oxygen Secies (ROS) production from Complex I, especially in conditions of ischemia-reperfusion injury [49,50]. During ischemia, the low level of oxygen would be responsible for the complete reduction in the respiratory chain components, in particular, the high ratio CoQH_2_/CoQ would result in a concomitant increase in ROS production. Recently, the role of the redox state of Coenzyme Q was reappraised, confirming the crucial function of the ratio of CoQH_2_/CoQ as an endogenous sensor for the mitochondrial function [51,52]. Low oxygen levels or impairment of Complex III/Complex IV would result in an excess of CoQ reduced form, due to an unbalancing between the quinone reducing enzymes activity and the quinol oxidizing enzymes. Excess reduced quinone would induce reverse electron transfer by stimulating ROS production from Complex I, which in turn would cause a partial degradation of mitochondrial Complex I itself. The purpose of this vicious cycle is to avoid excessive ROS production that would be deleterious to cell survival.

Besides Complex I and II, several dehydrogenases, mostly FAD-dependent, can fuel electrons to the ubiquinone pool, for example, ETF-dehydrogenase, involved in fatty acid beta-oxidation, or dihydroorotate dehydrogenase, which is essential for the de novo synthesis of pyrimidine nucleotide. In addition to the above, glycerol-3-phosphate dehydrogenase, which connects glycolysis and fatty acid metabolism with the oxidative phosphorylation [53], and proline and choline dehydrogenases, which use CoQ as electron acceptor, are included [54,55] (Figure 2). Finally, a quinone dependent D-lactate dehydrogenase has been recognized to be over-expressed in cancer cells [56], where it contributes to the detoxification of the methylglyoxal, a toxic metabolite derived by aberrant glucose catabolism and associated with cancer progression and Alzheimer’s disease [57,58]. A metabolic switch from glucose to fatty acid metabolism affects electron entry into the respiratory chain, favoring the activity of flavin-dependent dehydrogenases over NADH-dependent dehydrogenases. This will lead to an excess of FADH_2_ oxidation with a concomitant increase in the reduced state of the CoQ pool, which is considered responsible for the reverse electron transfer and the increased ROS production from mitochondrial Complex I. Szibor M et al. demonstrated that the expression of an alternative oxidase (AOX) could oxidize the quinone pool, releasing the excess of reduced quinone and inducing a shift from RET to FET (Forward Electron Transfer) [59].

### 2.2. CoQ10 in Cell Membranes

In non-mitochondrial biological membranes, CoQ continuously switches between reduced and oxidized forms through the action of enzymes with CoQ reductase activity [60]. These enzymes encompass NAD(P)H dehydrogenases, which are part of the plasma membrane redox system (PMRS). Among these reductases, CytB_5_R_3_ [61] and NQO1 [62] are the most important enzymes which maintain ubiquinol levels in cell membranes [63]. The PMRS becomes essential for the maintenance of bioenergetics when the mitochondrial activity is reduced, such as in aging. PMRS upregulation has been demonstrated in mtDNA-deficient cell lines with cytosolic NADH accumulation. Through NADH re-oxidation, PMRS allows cells to maintain the correct NAD+/NADH ratio required to sustain glycolytic flux [21]. 

In a recent paper [64], we demonstrated that in a human glioma cell line (T67) treated with 4-NB, a competitive inhibitor of the COQ2 enzyme, the endogenous level of CoQ10 decreased by 50%. This CoQ depletion, in addition to a decrease in the mitochondrial respiratory chain activity, induces a decrease also in the PMRS activity, confirming the crucial role of CoQ in maintaining this extra-mitochondrial electron transfer activity. Our results showed that, although the cells try to counteract the mitochondrial dysfunction upregulating the PMRS enzymes, the CoQ10 depletion induced a strong decrease in the efficiency of the plasma membrane redox system. Exogenous CoQ supplementation induces a strong increase in the cellular ubiquinone content, restores the PMRS efficiency, and increases the mitochondrial oxygen consumption rates. Although CoQ supplementation increased the PMRS efficiency over control levels, it was not able to completely restore the mitochondrial oxygen consumption rate, suggesting the existence of a rate-limiting step in the intracellular distribution for a so strongly hydrophobic molecule.

Besides its role in the redox state regulation, together with PMRS activity, CoQ plays an important role as a lipid-soluble antioxidant, preventing lipid peroxidation in biological membranes. It represents the only endogenously synthesized lipid-soluble antioxidant capable of protecting lipids, proteins, and DNA from oxidative damage [16]. Unlike other antioxidants, ubiquinol can inhibit both the initiation and propagation of oxidative damage, preventing the formation of lipid peroxyl radicals (LOO^•^); it reacts with preferryl radical and radicals generating ubisemiquinone and a non-radical lipid hydroperoxide [22,65]. Moreover, UQH_2_ effectively regenerates vitamin E from the α-tocopheroxyl radical, which additionally contributes to slow the propagation step of lipid peroxidation.

The exceptionally high antioxidant efficiency of CoQ is due to its intra-membrane localization, its general and abundant distribution, and its effective reduction/reactivation by several cellular systems that catalyze its reduction to the active form [16]. In mitochondria, the antioxidant reduced form of CoQ is regenerated directly by the respiratory chain [19]. In addition to the mitochondrial respiratory chain, the PMRS also exerts the same function, generating ubiquinol by transferring two electrons in a two single-step mechanism (CytB5R3) [61] or by direct two-electrons quinone reduction (NQO1) [62]; this last mechanism avoids the formation of a semiquinone intermediate (Figure 3). 

Mitochondria also play a role in ferroptosis, a unique mode of cell death, driven by iron-dependent phospholipid peroxidation. Ferroptosis is regulated by multiple cellular metabolic pathways; the cyst(e)ine–GSH–GPX4 axis is considered the main system opposing ferroptosis in mammals [66]. Recently, the CoQ oxidoreductase ferroptosis suppressor protein 1 (FSP1) has been shown to inhibit ferroptosis together with the glutathione peroxidase 4 (GPX4) [67]. Flavoprotein AIFM2 (apoptosis-inducing factor mitochondria-associated 2), previously characterized as a pro-apoptotic and P53-responsive protein associated with the mitochondrial outer membrane, is now considered as a novel ferroptosis modulator and renamed FSP1 (ferroptosis suppressor protein 1). Myristoylated FSP1 is associated with several cell membrane structures, including the Golgi apparatus, perinuclear structures, and the plasma membrane, where it serves as an NADPH-dependent oxidoreductase to mediate the reduction of ubiquinone and trap lipid peroxyl radicals, thus suppressing lipid peroxidation [66] (Figure 3). In a recent paper, we demonstrated that, in cultured cells, exogenous CoQ supplementation can protect membrane lipids from peroxidation and increase the resistance of cells to ferroptotic stimuli, such as treatment with erastin or RLS3, an inhibitor of GPX4 [68]. Interestingly, ferroptosis is associated with several neurodegenerative disorders (Parkinson’s and Alzheimer’s diseases), chronic inflammation, frailty, and cancer; therefore, CoQ10 supplementation could be beneficial to counteract the age-related progression of these diseases [69].

### 2.3. CoQ as a Component in Low Density Lipoproteins (LDL)

Besides its cellular involvement in bioenergetic functions, CoQ is also a component of LDL; this suggests its potential role in atherosclerotic prevention [70,71]. The mode of action in protecting the LDL from oxidation is similar to the one described for cell membranes, as previously discussed, together with α tocopherol [71]. CoQ and α tocopherol protect from cellular damage in proportion to the amount of their content in the LDL membrane, suggesting that supplementation with these molecules could overcome LDL oxidation.

Interestingly, HepG2 cells can reduce extracellular CoQ10 to CoQ10H_2_ using reductase present on the outer surface of cells, providing reduced levels of CoQ10 in plasma [72]. Moreover, CoQ10, vitamin E, and dihydro thioctic acid can cooperatively act to prevent LDL oxidation [73].

The double-edged knife of cardiovascular risk prevention is well described by the negative action of pharmacological prevention with statins on physiological CoQ10 levels. Statins block endogenous production of cholesterol in the liver by inhibiting 3-hydroxy-3-methylglutaryl-Coenzyme A (HMG-CoA) reductase [74], affecting CoQ synthesis and thus accelerating CoQ depletion in elderly people. 

The negative effects of long-term utilization of statin, in the primary and secondary intervention of lowering cardiovascular risk, are evident mostly in muscle and liver in selected groups of elderly people. The reason for such side effects could reside in genetic predisposition. The gene *SLCO1B1* (solute carrier organic anion transporter family member 1B1) codes for an organic anion-transporting polypeptide that is involved in the regulation of the absorption of statins. A common variation in this gene was found to significantly increase the risk of myopathy [75].

### 2.4. Aging and Longevity: Role of CoQ10 Homeostasis

Exercise and oxygen consumption through the respiratory chain usually produce free radicals in the form of ROS (Reactive Oxygen Species), which participate in regular hormetic response [76]. The hormetic response consists of, among other things, a major production of antioxidant enzymes such as superoxide dismutase (SOD), catalase, and glutathione peroxidase (GPx). Furthermore, to protect macromolecules from oxidative damage, cells contain antioxidant molecules such as: ascorbic acid, alpha-tocopherol, glutathione, and coenzyme Q10 (CoQ10), which are used by the antioxidant enzymes and other systems, such as cytochrome b5 reductase (CytB_5_R_3_) or NAD(P)H quinone dehydrogenase 1 (NQO1). Currently, it has been established that aging is associated with an accumulation of malfunctioning mitochondria at the cellular and tissue level [77,78,79]. Because of its central role in mitochondrial physiology, a decrease in CoQ10 levels could contribute to accelerating the dysfunction of mitochondrial activity associated with aging and related diseases [80]. CoQ10 is also a potent antioxidant at the level of the cellular plasma membrane and in plasma lipoproteins. The reduced form of CoQ10, ubiquinol (UQH_2_), prevents both the initiation and propagation of lipid peroxidation in cell membranes [81] and plasma lipoproteins. CoQ10 also regulates gene expression, especially of genes involved in the chronic inflammatory response, and exhibits anti-inflammatory properties [15,82,83]. Therefore, it seems clear that antioxidant integration is necessary to counteract the negative effects of age-related ROS accumulation. The central physiological role of CoQ10 points to it as the best antioxidant to supplement. CoQ10 is also involved in several cellular functions that should be studied in detail to target an effective supplementation intervention for aging and age-related disorders.

### 2.5. CoQ in Human Diseases

In human disease, CoQ10 deficiency has been described both as a primary and secondary defect and has been involved in the pathogenesis of different conditions (Figure 4). Primary CoQ10 deficiencies are rare autosomal recessive disorders that can affect many organs and tissues, especially the brain, muscles, and kidneys, involving a shortage (deficiency) of CoQ10 due to mutations in the genes involved in the biosynthesis of CoQ10. Secondary CoQ10 deficiencies are caused by defects in mitochondrial and non-mitochondrial processes associated with a secondary reduction in CoQ10 in cells or tissues. 

Different clinical phenotypes can be strictly related to CoQ10 deficiency at a specific tissue level. Primary CoQ10 deficiencies are characterized by five major phenotypes due to mutations in any of the genes involved in CoQ10 biosynthesis, namely encephalomyopathy, cerebellar ataxia, infantile multisystemic form, isolated myopathy, and nephropathy [84]. Up to now, mutations in *PDSS1*, *PDSS2*, *COQ2*, *COQ4*, *COQ5*, *COQ6*, *COQ7*, *COQ8A*, *COQ8B*, and *COQ9* genes have been found to cause human CoQ deficiency disease, but only about 280 patients from 180 families have been reported in the literature.

The extreme variability in clinical and phenotypic presentation, such as the age of onset, severity of disease, and organs involved, make the existence of a unique common pathway for all the forms unlikely. Thus, recently, the presence of a possible unknown function of CoQ10 genes not strictly related to oxidative phosphorylation (OXPHOS) function is starting to be postulated. A therapy with CoQ10 supplementation is actually present in the clinical practice. Although this compound is used and could give good therapeutic prospects, no clinical trials or long-term phase III studies have yet been performed to truly document and prove the efficacy of this compound in either primary or secondary CoQ10 deficiency. In the primary deficit of CoQ10 such clinical trials might be difficult to implement, first because of the very low number of patients, and also because of the severity of the diseases, the limited data on the natural history of these diseases, and, ultimately, the few outcome measures available.

Quinzii et al. described an accumulation of hydrogen sulfide (H_2_S) in primary CoQ deficiencies, emphasizing one specific role of CoQ10 with the participation of the sulfide oxidation pathway via sulfide:quinone oxidoreductase (SQOR) [85]. The close relationship between H_2_S and CoQ levels can explain these findings in primary CoQ disorders, in which H_2_S is formed by desulfuration of homocysteine and cysteine via Cystathionine-β synthase (CBS) and cystathionine γ lyase (CSE). High levels of H_2_S seem to have deleterious effects as COX inhibitors, while at the physiological level can be utilized by the Krebs cycle as a substrate. Unbalanced homocysteine levels, in turn, can decrease H_2_S levels [86], as observed in hyperhomocysteinemia related to cardiovascular diseases. Regulation of H_2_S levels can thus provide beneficial effects both in primary and secondary CoQ10 deficiencies, becoming a key point in therapeutic approaches.

CoQ10 deficiency is much more common in association with other diseases, either related to a specific genetic condition not directly involved in CoQ10 biosynthesis or in nongenetic disorders as a side effect. Secondary CoQ10 deficiency is often present in mitochondrial disease in which CoQ10 biosynthesis genes are not directly involved, but with a clinical presentation similar to the ones described in many mitochondrial diseases, such as cardiomyopathy or myopathy, exercise intolerance, or brain diseases and epilepsy. In these diseases, the administration of a high dose of CoQ10 is a common therapy even if a clear efficacy has not yet been proved. Moreover, a deficit in CoQ10 has been identified in some genetic disorders linked to lipid metabolism, such as Familial Hypercholesterolemia and MAAD (Multiple acyl-CoA Dehydrogenase Deficiency) [87]. In both these conditions, the administration of CoQ10 lead to a reduction in the atherosclerosis and to a reduction in muscle pain and fatigue. More recently, a role in CoQ10 in inflammation and in autoimmune diseases has been postulated, with it being proven that CoQ10 supplementation reduced the levels of circulating inflammatory markers. This CoQ10 anti-inflammatory potential might be related to the downregulation of genes regulated by nuclear factor-κβ (NF-κβ), which is known to be activated by reactive oxygen species [88].

The role of CoQ10 has been described in fibromyalgia, a chronic pain syndrome whose pathophysiological mechanisms have not yet been identified, and a decrease in mitochondrial mass and CoQ10 levels, as well as an overproduction of ROS, was detected in blood mononuclear cells from patients affected by fibromyalgia [89]. Moreover, the pain in fibromyalgia could also be mediated by the activation of the inflammasome by CoQ10 [90]. The role of CoQ10 in inflammation could be in part responsible for some secondary mitochondrial dysfunction, and CoQ depletion is also present in different chronic age-related disorders such as type 2 diabetes [91], with or without insulin resistance, cardiovascular disease and atherosclerosis [92], neurodegeneration, in particular Multiple System Atrophy [93], chronic kidney disease [94], liver disease [95], and sarcopenia [96]. 

## 3. Coenzyme Q Determination in Biological Samples

### 3.1. CoQ Determination

The gold standard procedures for CoQ10 determination in biological samples are based on high-pressure liquid chromatography. Commonly, the ultraviolet detection system (HPLC-UV) is used for total ubiquinone quantification [97,98], while the electrochemical (HPLC-ED) detection system [99,100,101,102], by performing reduction or oxidation in the electrochemical cell after chromatographic separation, allows investigating the CoQ redox status. In addition, liquid chromatography–mass spectrometry (LC–MS) and liquid chromatography coupled to tandem mass spectrometry (LC–MS/MS) methods have been developed for the analysis of CoQ in tissues due to their high sensitivity and selectivity [103]. The quantification of the reduced (ubiquinol) and oxidized (ubiquinone) forms is more complicated than the determination of total CoQ, since ubiquinol is particularly sensitive to oxidation; this makes it necessary that the sample be immediately frozen at −80 °C after collection [104,105,106]. Taking this critical issue into account, the ubiquinol/ubiquinone ratio has been used as a marker of oxidative stress [99,107], for example in patients with hyperlipidemia or liver disease [101], and to characterize the level of ubiquinone deficiency in pediatric patients [108]. In addition to the stability problems of the reduced form of CoQ, one of the issues in the analysis of Coenzyme Q in biological samples is the lack of an agreed, shared internal standard. In most cases, Coenzyme Q9 is the internal standard of choice [106]; nevertheless, it is affected by contamination from diet and synthesis by intestinal microorganisms, contributing up to 2–7% of the total ubiquinone pool [109]. Several non-physiological analogs (CoQ6, CoQ7, CoQ11) [101,105,110,111], di-propoxy- [112], and diethoxy- CoQ10 [113] have been used to avoid the possible influence of endogenous CoQ analogs in the determination of ubiquinone content.

### 3.2. Biological Samples

Plasma content is widely used in the clinical assessment of CoQ10 deficiency. The established reference range for plasma CoQ10 status varies from approximately 0.26 to 1.7 μM, and this wide range of values suggests that plasma CoQ content may be affected by multiple factors [104,112,114].

Plasma CoQ determination, being minimally invasive, is particularly useful in monitoring CoQ levels in patients following supplementation [115]; follow-up of patients treated with exogenous CoQ should include periodic quantification of plasma CoQ for dose adjustment, to monitor treatment compliance, and to assess adequate intestinal absorption of CoQ [116]. However, the degree and mode of distribution of plasma CoQ to tissues is not completely known and is still a matter of debate.

Plasma levels of CoQ are affected by dietary intake, contributing up to 25% of the total amount; although CoQ has a relatively long circulatory half-life (approximately 24 h), its bioavailability from the diet is generally low [109,112]. Beyond the dietary intake, hepatic biosynthesis affects the plasma CoQ level. This situation does not mirror other tissues, which are dependent upon de novo biosynthesis [117,118]. Finally, being 58% of total plasma CoQ associated with low-density lipoprotein (LDL) fraction [119,120], LDL concentration may also influence plasma CoQ content. Therefore, in view of its dependence upon dietary intake, biosynthesis, and lipoprotein concentration, plasma CoQ status may not truly reflect tissue levels. As reported by some authors, there is no correlation between plasma CoQ concentrations and skeletal muscle or mononuclear cells (MNC) [112,121] or platelets [122]. There is no agreement in the literature as to whether or not gender and race affects plasma CoQ content; some authors report a significant difference [105,123] while others report no difference [104,124,125]. On the other hand, plasma CoQ’s redox state seems independent of sex and racial differences [126].

Similar to gender, there is no agreement in the literature as to whether age affects the plasma amount of CoQ. Duncan et al. [112] reported that aging had no significant influence on CoQ10 concentrations in the reference population. In contrast, other authors reported that separate reference intervals for total CoQ10 according to age are justified [99,105,123].

A strong correlation was found for total CoQ and cholesterol concentrations, and the correlation of CoQ10 with age disappeared when cholesterol was included in a multivariate analysis [99,123]. For these reasons, it has been suggested that plasma CoQ10 levels should be normalized on plasma cholesterol or LDL fraction [127,128,129]. However, even following these normalizations, in many patients with primary CoQ deficiency plasma values are normal [116]. This represents a limitation in the use of this biological sample for the diagnosis of these diseases, and whether plasma CoQ can be used to indicate deficient tissue CoQ status in patients with mitochondrial disorders remains unknown at present [116]. Since a decrease in CoQ plasma concentration is a rare condition, it might entail a secondary CoQ deficiency, such as phenylketonuria and lysosomal storage diseases [130,131].

As an alternative to plasma, the use of blood cells has been suggested to determine endogenous CoQ content. Mononuclear cells (MNCs) are easily isolated from whole blood, and their CoQ content correlates with that of skeletal muscle and is affected by exogenous supplementation [21,112,132,133]. Since platelets contain mitochondria, they have been used to assess endogenous CoQ content and to monitor the effect of exogenous CoQ supplementation in several clinical trials [122,134,135,136]. In addition, platelet CoQ content is an index of mitochondrial electron transport chain function [102,137].

The most prominent limitation of using blood cells, although they represent a useful model for determining intracellular CoQ concentrations, is the lack of shared reference values and detailed information regarding sample preparation and detection methods [116,122].

Fibroblasts have also been used for the diagnosis of CoQ deficiency, and several ranges of their ubiquinone content have been reported. However, the tissue specificity of CoQ deficiencies does not exclude that a normal level of CoQ in fibroblasts may not reflect a deficiency in other tissues [138]. These cells are a suitable model for studying the ubiquinone biosynthetic pathway [28,30] and for discriminating between primary and secondary deficiencies.

Although it is an invasive assay, skeletal muscle has been considered the tissue of choice for determination of ubiquinone levels since the first cases reported by Ogasahara et al. [138]. Muscle CoQ is not affected by age, gender, and ethnicity and, therefore, is a good indicator of CoQ deficiency, and it has been proposed that CoQ level may be an indicator of mitochondrial respiratory chain impairment [139,140].

Indeed, several authors report a strong correlation between CoQ levels and the integrated activity of Complexes II and III of the mitochondrial respiratory chain. However, it has been suggested that in order to prevent possible errors in the diagnosis of CoQ deficiency, the amount of muscle CoQ should be normalized to both citrate synthase activity and total protein content [141].

## 4. Coenzyme Q10 Supplementation

All human nucleated cells are capable of synthesizing CoQ10, but an exogenous amount (3 to 5 mg/day) is absorbed from dietary sources [142]. Due to its physicochemical similarity with Vitamin E, CoQ10 should follow the same route of absorption of the latter (chylomicron remnants, liver uptake), where ubiquinol is repackaged into low density lipoproteins and then rereleased into the circulation. In both primary or secondary CoQ10 deficiencies, a therapeutic approach using CoQ10 as a supplement has been proposed both for its qualities as an antioxidant agent and for its ability to ameliorate the OXPHOS function. Unfortunately, ubiquinol and ubiquinone have a very low bioavailability due to poor water solubility, instability to light, and thermolability. The efficiency of absorption of CoQ10 formulations seems to decrease as the dosage increases, with a suggested block of GI absorption above 2400 mg, and dietary fat together with grapefruit juice consumption have been reported to improve the absorption of CoQ10 [143]. To date, there is no information on an optimal endogenous therapeutic range for CoQ10 after exogenous supplementation. Some studies suggest oral supplementation doses of up to 2400 mg per day in adults and up to 30 mg/kg per day in children. Although there are no data on therapeutic concentrations of CoQ10 in patients with CoQ10 deficiency, a study in patients with congestive heart failure reported that a blood CoQ10 concentration of 4.1 μM was therapeutically effective. Several studies have suggested that blood mononuclear cells may provide the best data for monitoring endogenous CoQ10 compared with plasma because of a strong correlation between CoQ10 levels in skeletal muscle and blood mononuclear cells, but not with plasma in patients without evidence of disease related to mitochondrial function [144]. Regarding oral intake, different physiological factors influence the bioavailability of compounds, e.g., sex, age, genetic phenotype, the health of the gastrointestinal tract, various health disorders, the administration route, and interactions with food. Such difficulties in reaching the therapeutic levels of this molecule drive the efforts in designing new preparations for oral intake because of wide use of CoQ10 as a complementation approach in older individuals. Pravst and colleagues [145] showed that a water-soluble CoQ10 syrup preparation had a 2.4-fold higher bioavailability with respect to a ubiquinone capsule preparation, even if CoQ10 appeared in the blood mostly as ubiquinol and even if consumed as ubiquinone. Such limitations in bioavailability compromise its efficacy and, consequently, its use and efficacy as a drug in human diseases. Hence, several formulations have been developed to face such inconvenience. The most promising strategy to overcome this low bioavailability has been to encapsulate CoQ10 in lipid nanoparticles, as they can overpass through biological membranes, and eventually evolved towards chemical modifications of the molecule to decrease its hydrophobicity. The aim of this strategy is to improve stability, prolong circulation times, and increase the bioavailability of CoQ10 [146]. A similar approach is represented by the self-nanoemulsifying delivery system, an isotropic and thermodynamically stable mixture of an oil, a surfactant, a co-surfactant, and a drug that forms nanoemulsion droplets of a reduced size. Nevertheless, to avoid the negative impact on the efficacy of lipid-based delivery methods, such as degree of emulsification, particle size, or drug precipitation, a novel lipid-free nanoformulation has been tested using various surfactants, but no other lipids. These nano-CoQ10 particles were modified with surfactants using hot high-pressure homogenization (HPH) [147]. 

Masotta et al. found an improvement in the bioavailability using CoQ10-loaded oleogels, an emulsion where CoQ 10 is dissolved in an oil-dispersed phase [148]. Lastly, a water-soluble CoQ10 has been identify where CoQ10’s bioavailability has been increased by encapsulating it in β-cyclodextrin inclusion complexes [149]. This novel patented formulation has proved to be stable and well-soluble in diverse aqueous media.

## 5. Conclusions

The central role of Coenzyme Q in the mitochondrial energy transducing system is well known. Nevertheless, CoQ has several functions in cell metabolism, and a decrease in total cellular CoQ concentration is responsible for the impairment of several vital functions.

CoQ deficiencies are classified as primary, associated with genetic modification of genes involved in CoQ biosynthesis, and secondary, depending on other diseases or associated with pharmacotherapies and with the activation of inflammasome. Exogenous CoQ supplementation can ameliorate the clinical status in patients with CoQ deficiency. Nevertheless, in many patients, CoQ therapy is not effective; it is calculated that only 20% of patients are responsive to exogenous CoQ10 [150]. The lack of effectiveness of CoQ therapy may depend on several factors that can be associated both with the low bioavailability of this very hydrophobic molecule and the broad series of metabolic modifications induced by decreased levels of CoQ. 

While the low bioavailability of the molecule can be improved through the development of more efficient formulations, the study of the pathological implications due to ubiquinone depletion appears extremely complicated.

There are conflicting data in the literature regarding ubiquinone levels measured in tissue biopsies or plasma samples, and the quantification and normalization of such data are subject to often unresolvable experimental limitations. In primary ubiquinone deficiencies, the effect of mutations directly involved in the biosynthetic pathway of the molecule appears clear and is reflected in a direct depletion of the molecule. On the other hand, concerning secondary deficiencies, it is difficult to even classify them based on their main pathological effects. 

The extreme heterogeneity of pathologies characterized by ubiquinone deficiency suggests that the molecule is involved in multiple, often interacting, cellular mechanisms. The lack of knowledge of these mechanism makes the relationships between cause and effect ambiguous and hard to understand.

Moreover, this makes it impossible to recognize pathologies that could be classified as secondary deficiencies and that could benefit from CoQ supplementation. CoQ supports cellular function through the maintenance of the electron transport chain, protection against free radicals, and support of lysosomal function; in general, regarding the electron transport chain, CoQ deficiencies have been associated with reduced activity of complex II and III and increased activity of complex I [151]. Therefore, a question arises regarding CoQ deficiency: is it the cause of mitochondrial dysfunction or is it caused by the latter condition?

For these reasons, the study of the basic cellular mechanisms involving CoQ, combined with genetic and clinical data, may contribute to the development of effective therapies in pathologies in which CoQ10 deficiency represents a well-characterized feature, thus facilitating the treatment of these pathologies, regardless of their etiopathogenesis.

## Figures and Tables

**Figure 1 ijms-23-00128-f001:**
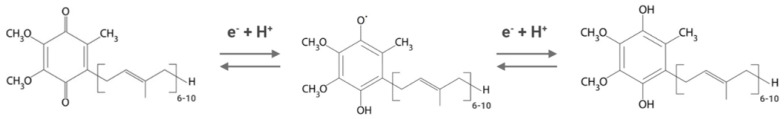
Different redox states of Coenzyme Q. From left to right: oxidized form (ubiquinone), semi-reduced form (semiquinone), and fully reduced form (ubiquinol). Depending on the organism, the isoprenoid chain is composed of 6 to 10 units.

**Figure 2 ijms-23-00128-f002:**
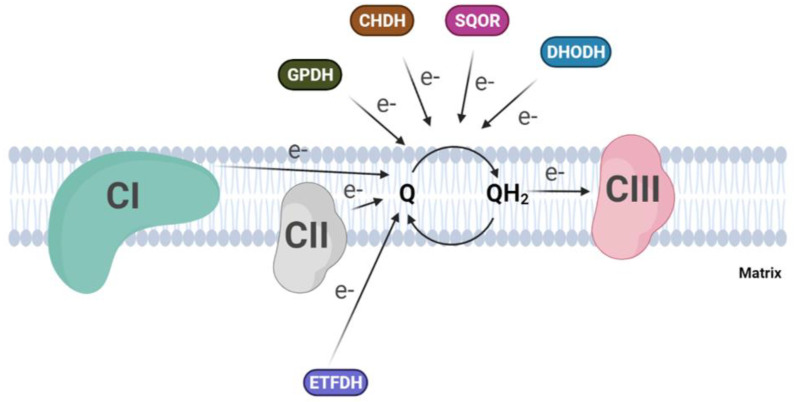
Convergent electron flow to the Coenzyme Q pool in the inner mitochondrial membrane. (CI, Complex I; CII, Complex II; CIII, Complex III; GPDH, Glycerol-3-phosphate dehydrogenase; CHDH, choline dehydrogenase; SQOR, sulphide:quinone oxidoreductase; DHODH dihydroorotate dehydrogenase).

**Figure 3 ijms-23-00128-f003:**
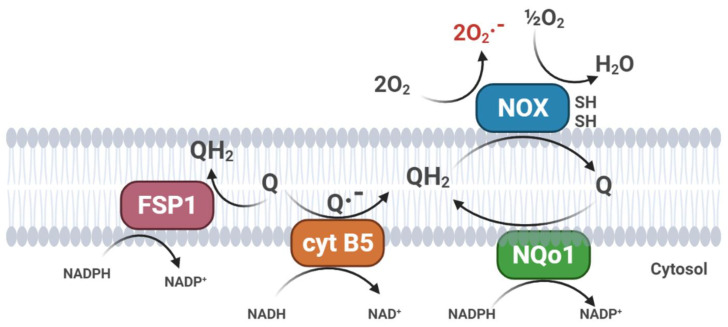
Plasma membrane redox systems (PMRS). (FSP1, Ferroptosis Suppressor Protein 1; cyt B5, cytochrome b5 reductase; NQo1, NAD(P)H: quinone oxidoreductase; NOX, plasma membrane NADH oxidase).

**Figure 4 ijms-23-00128-f004:**
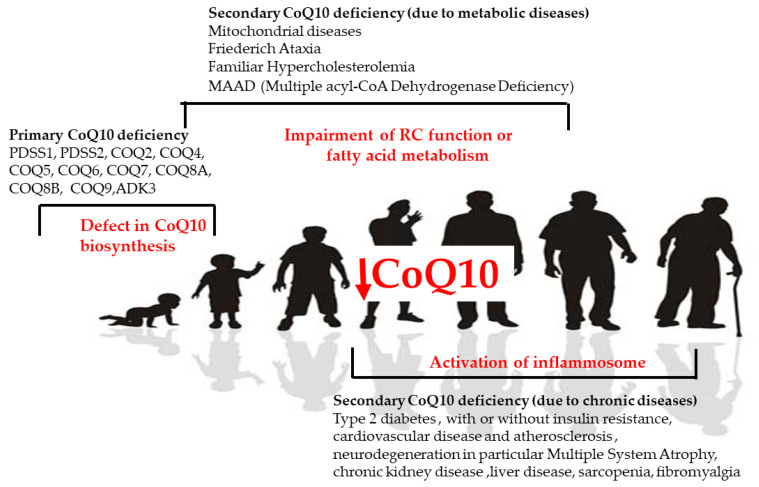
CoQ in human diseases. Causes of primary and secondary deficiency of CoQ10 in human disease.

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
