# Peer review of "The Roles of Coenzyme Q in Disease: Direct and Indirect Involvement in Cellular Functions"

_ijms, 2021, doi:10.3390/ijms23010128_

Round 1

Reviewer 1 Report

The manuscript by Pallotti et al provides a good overview of the current knowledge around CoQ10 and its association with some pathological states to make a case for the development of CoQ-dependent therapies. Overall, the manuscript is well written and leads the reader through the topic in a systematic and intuitive way. There are some repeats (i.e. CoQ10 is a potent antioxidant) that could have been avoided to shorten the manuscript and some important aspects were either ignored or provided in a manner to suit the overall hypothesis.

One limitations of the manuscript is a lack of discussion around the different techniques to measure CoQ10 that makes most in vivo and in vitro studies incomparable and should be the foundation of our understanding. How can we even discuss deficiency disorders and effective supplementation if we cannot agree on a common and accurate method to measure CoQ10 levels?

The second limitation relates to the chemical reduction of CoQ10. The authors put forward two enzymes (reductases) CytB5R3 and NQO1. At least for NQO1 the literature that is reduces CoQ10 at physiological relevant rates, which could explain its antioxidant activity is more than shaky. Moreover, as a purely cytoplasmic protein it is unclear how it could even interact with the quinone head, which, based on the available modelling, is situated inside the membrane.

Another omission was the discussion around reduction substrates. While it is clear that NADH is the electron donor in the mitochondrial ETC, in the cytoplasm the situation is very different. In the cytoplasm the majority of NADH/NAD+ is present in the oxidised form as NAD+ and hence can not be used to reduce CoQ10 (Williamson DH, Lund P, Krebs HA (1967) Biochem. J103 (2): 514–27. ). On the other hand, for NADPH/NADP+ the majority is in the reduced state (Veech RL, Eggleston LV, Krebs HA (1969) Biochem. J115 (4): 609–19) and could act as electron source. Moreover, NQO1 as cytoplasmic enzyme prefers NADPH as substrate over NADH. Since one of the main functions of PPP produced NADPH is to reduce GSSG to GSH, CoQ10 in this context would appear as just another cellular antioxidant that is activated by this mechanism. Therefore, a more detailed representation of the enzymes and associated substrates in different compartments would help the manuscript.

Irrespective of reduction sites and reduction substrates, the discussion around the reduction states of CoQ10 is unclear. Chemically semiquinones are highly unstable molecules that donate the excess electrons preferentially to oxygen to produce superoxide. Hence uncontrolled single electron reduction is avoided as much as possible by 1. Use of a 2-electron reduction or 2. Two fast 1-electron reduction steps, as thought to happen withing complex I. Therefore, some statements such as “… generating ubiquinol through one-electron (CytB5R3) reductions…” are clearly wrong. This type of reduction would also not generate an antioxidant molecule but a prooxidant (BTW, this is also a topic that was not discussed as there is literature on the pro-oxidant function of CoQ10. Any antioxidant can function as a pro-oxidant) and could certainly be an issue for high doses of dietary CoQ10.

In this context, a table or similar to show the reducing systems in the different cellular compartments would be helpful (or if not available, this lack of information should also be mentioned for a balanced presentation of the current state of knowledge). How for example is CoQ10 reduced in the nuclear membrane? It is clear from in vitro experiments that lateral membrane diffusion of CoQ10 is a VERY slow process. In cells from CoQ10-deficiency syndrome patients it took 1 week of continuous exposure to CoQ10 to restore mitochondrial deficits (Lopez et al. 2010). Given that this system did not include CoQ10 metabolism (which was also not discussed in the manuscript) that is seen in vivo systems, it begs the question if dietary CoQ10 could be transported fast enough to target organelles to provide a therapeutic impact at all.

Given that CoQ10 supplementation is somewhat effective in only about 20% of patients in CoQ10 deficient patients (which should be the perfect scenario to show any activity at all), any discussion about therapeutic use of CoQ10 in non CoQ10 deficiency syndromes should be backed up by serious clinical data such as from properly powered and controlled clinical phase III trials or Cochrane reviews to put the discussion on an evidence-based footing instead of citing underpowered, short-term studies with high risk of bias. If the evidence is not convincing enough, this should be also stated to highlight what we need to focus on in future studies.

Some proofreading might be necessary. For example, in the Introduction: “… is the most ancient way to starve energy in biological systems [9].” should probably read “store energy”.

Author Response

Cover letter for the reviewer 1 of manuscript ijms-1461538

Dear Editor,

We would like to thank both reviewers for their helpful suggestions that lead us to modify (and hopefully improving) this present final version of the article.

Going through a point-by-point analysis, we modified the article following reviewers’ criticisms.

REVIEWER 1

One limitations of the manuscript is a lack of discussion around the different techniques to measure CoQ10 that makes most in vivo and in vitro studies incomparable and should be the foundation of our understanding. How can we even discuss deficiency disorders and effective supplementation if we cannot agree on a common and accurate method to measure CoQ10 levels?

We added (highlighted in yellow) a new section on “CoQ determination in biological samples” presenting a discussion on the different methods used for the determination of CoQ levels in biological samples.

The second limitation relates to the chemical reduction of CoQ10. The authors put forward two enzymes (reductases) CytB5R3 and NQO1. At least for NQO1 the literature that is reduces CoQ10 at physiological relevant rates, which could explain its antioxidant activity is more than shaky. Moreover, as a purely cytoplasmic protein it is unclear how it could even interact with the quinone head, which, based on the available modelling, is situated inside the membrane.

We agree with your comment. We only cite these reductases as ubiquinol generators. At least, NQO1 has been fully characterized and its role is essential as detoxifier in protecting plasma membrane form lipid peroxidation, being an excellent CoQ10 reducer in lipid environment, as cited in Ross D, Siegel D. The diverse functionality of NQO1 and its roles in redox control. Redox Biol. 2021 May;41:101950. We therefore changed Ref 62 with this new review.

Some proofreading might be necessary. For example, in the Introduction: “… is the most ancient way to starve energy in biological systems [9].” should probably read “store energy”.

We corrected as (pale blue highlighted) Indeed, the formations of trans-membrane ions gradients is the most ancient way to store energy in biological systems [9].

Another omission was the discussion around reduction substrates. While it is clear that NADH is the electron donor in the mitochondrial ETC, in the cytoplasm the situation is very different. In the cytoplasm the majority of NADH/NAD+ is present in the oxidised form as NAD+ and hence can not be used to reduce CoQ10 (Williamson DH, Lund P, Krebs HA (1967) Biochem. J103 (2): 514–27. ). On the other hand, for NADPH/NADP+ the majority is in the reduced state (Veech RL, Eggleston LV, Krebs HA (1969) Biochem. J115 (4): 609–19) and could act as electron source. Moreover, NQO1 as cytoplasmic enzyme prefers NADPH as substrate over NADH. Since one of the main functions of PPP produced NADPH is to reduce GSSG to GSH, CoQ10 in this context would appear as just another cellular antioxidant that is activated by this mechanism. Therefore, a more detailed representation of the enzymes and associated substrates in different compartments would help the manuscript.

We agree with the observations raised by the referee on the direct electron donor for the reduction of the CoQ present in non-mitochondrial membranes, nevertheless it is not easy in vivo to discriminate between NADH and NADPH as reduction substrate. In literature there are several pieces of evidence showing that, in vitro, NQO1 can use NADH to reduce long chain CoQ derivatives incorporated both in artificial and natural membranes.

Irrespective of reduction sites and reduction substrates, the discussion around the reduction states of CoQ10 is unclear. Chemically semiquinones are highly unstable molecules that donate the excess electrons preferentially to oxygen to produce superoxide. Hence uncontrolled single electron reduction is avoided as much as possible by 1. …This type of reduction would also not generate an antioxidant molecule but a prooxidant (BTW, this is also a topic that was not discussed as there is literature on the pro-oxidant function of CoQ10. Any antioxidant can function as a pro-oxidant) and could certainly be an issue for high doses of dietary CoQ10.

We completely agree that an antioxidant molecule can be prooxidant at high doses; however, to our knowledge, there is no evidence for prooxidant activity of plasma CoQ in vivo. On the other hand, there is evidence for prooxidant activity of CoQ in vitro at high doses (Bergamini C, Moruzzi N, Sblendido A, Lenaz G, Fato R. A water soluble CoQ10 formulation improves intracellular distribution and promotes mitochondrial respiration in cultured cells. PLoS One. 2012;7(3):e33712).

Use of a 2-electron reduction or 2. Two fast 1-electron reduction steps, as thought to happen withing complex I. Therefore, some statements such as “… generating ubiquinol through one-electron (CytB5R3) reductions…” are clearly wrong

In reality here it is not intended that cyt b5 reductase gives only one electron to ubiquinone but that the reduction occurs by successive addition of one electron at a time, as corrected in highlighted green in the text.

How for example is CoQ10 reduced in the nuclear membrane? It is clear from in vitro experiments that lateral membrane diffusion of CoQ10 is a VERY slow process. In cells from CoQ10-deficiency syndrome patients it took 1 week of continuous exposure to CoQ10 to restore mitochondrial deficits (Lopez et al. 2010). Given that this system did not include CoQ10 metabolism (which was also not discussed in the manuscript) that is seen in vivo systems, it begs the question if dietary CoQ10 could be transported fast enough to target organelles to provide a therapeutic impact at all.

Lateral diffusion of CoQ is fast enough not to be rate-limiting for electron transfer in the mitochondrial transport chain, but the intracellular distribution of CoQ is a poorly understood process and detailed knowledge of possible transport proteins is still lacking. (Kemmerer ZA, Robinson KP, Schmitz JM, Manicki M, Paulson BR, Jochem A, Hutchins PD, Coon JJ, Pagliarini DJ. UbiB proteins regulate cellular CoQ distribution in Saccharomyces cerevisiae. Nat Commun. 2021 Aug 6;12(1):4769).However, exogenous administration of CoQ is able to increase mitochondrial ubiquinone content in vitro and improve mitochondrial activities in vivo. Whether exogenous CoQ10 can be transported fast enough to mitochondria to provide therapeutic impact is still an open question although there is evidence of improved mitochondrial parameters especially in primary CoQ deficiencies.

Given that CoQ10 supplementation is somewhat effective in only about 20% of patients in CoQ10 deficient patients (which should be the perfect scenario to show any activity at all), any discussion about therapeutic use of CoQ10 in non CoQ10 deficiency syndromes should be backed up by serious clinical data such as from properly powered and controlled clinical phase III trials or Cochrane reviews to put the discussion on an evidence-based footing instead of citing underpowered, short-term studies with high risk of bias. If the evidence is not convincing enough, this should be also stated to highlight what we need to focus on in future studies.

We improved the discussion about CoQ10 supplementation by adding a new paragraph and we reported the lack of clinical data in grey highlighted sections “Coq in human disease” and “Conclusion”

Reviewer 2 Report

 This is a nice review describing the early findings of CoQ, its role in the electron transfer system, roles in other metabolic processes, disease relation, ageing and an inhibitor.

The authors used the term CoQ and ubiquinone. They should better keep the term ‘CoQ’ in the whole text to be consistent. ‘ubiquinone’ ‘coenzyme Q’ ‘COQ’ and ‘coenzyme q’ should be ‘CoQ’. ‘ubiquinonol’ should be ‘CoQH2

Page 1,

The title ‘The roles of Coenzyme Q in disease: direct and indirect involvement in cellular functions’, expected us the disease-related subjects of CoQ, but the text is more broadly described about the role of CoQ and historical findings. It is difficult to follow the subject what the authors would like to explain from the title. The main parts are described in pages 7 and 8. I suggest shortening the content which is not so much relevant to the title, especially ‘Introduction’.

Pages, 1-4

I feel ‘Introduction’ is lengthy and overlapping with the latter parts. They spend three pages for ‘Introduction’. Some topics are repeatedly described in the main text. It is better to make Introduction more concise. Otherwise, it is not clear what is the main topics in this review

Page 2,

‘ubiquinones are present in all organisms’

Because ubiquinone is not present in gram positive bacteria, this statement is not accurate. Refer ‘Kawamukai, Biosci. Biochem. Biotechnol. 2018’.

Page 3,

Since the human genes for CoQ biosynthesis has been discovered as orthologues of the yeast genes, they should refer some works on yeasts such as ‘Hayashi et al PLoS one 2014’.

Page 4,

‘no qui-nones were found bound to respiratory enzymes, leading to speculation of a dynamic equilibrium between quinone pools.’

 This statement should be fixed. The CoQ binding site in Complex I has been structurally solved in several species (Zickermann et al. Science 2015; Yoga et al., Front. Chem. 2021)

Pieces of Evidence  >> Pieces of evidence

‘a D-lactate dehydrogenase quinone dependent is been recognized to be over-expressed’ >> a quinone dependent D-lactate dehydrogenase has been recognized to be over-expressed

Page 5,

CytB5R3[70], For the reference of Cytb5R3, ‘Villalba et al. Mol. Aspects Med. 1997’ is better.

Page 7,

The reference numbers should be re-organized orderly through the text. Especially, the order of the references from 60 to 85 should be fixed.

‘In addition to the few examples given at the beginning of this chapter,’

Is this chapter ?

Page 8,

‘some authors’ > Quinzii et al.,

Author Response

Cover letter for the reviewer 3 of manuscript ijms-1461538

Dear Editor,

We would like to thank both reviewers for their helpful suggestions that lead us to modify (and hopefully improving) this present final version of the article.

Going through a point-by-point analysis, we modified the article following reviewers’ criticisms.

REVIEWER 3

The authors used the term CoQ and ubiquinone. They should better keep the term ‘CoQ’ in the whole text to be consistent. ‘ubiquinone’ ‘coenzyme Q’ ‘COQ’ and ‘coenzyme q’ should be ‘CoQ’. ‘ubiquinonol’ should be ‘CoQH2

We modified as suggested

Page 1,

The title ‘The roles of Coenzyme Q in disease: direct and indirect involvement in cellular functions’, expected us the disease-related subjects of CoQ, but the text is more broadly described about the role of CoQ and historical findings. It is difficult to follow the subject what the authors would like to explain from the title. The main parts are described in pages 7 and 8. I suggest shortening the content which is not so much relevant to the title, especially ‘Introduction’.

We shortened the introduction section

Pages, 1-4

I feel ‘Introduction’ is lengthy and overlapping with the latter parts. They spend three pages for ‘Introduction’. Some topics are repeatedly described in the main text. It is better to make Introduction more concise. Otherwise, it is not clear what is the main topics in this review

We cut out part of the introduction by moving some parts in the section “Ubiquinone in mitochondrial electron transfer chains” and creating a new section entitled “Ubiquinone/Coenzyme Q biosynthesis”

Page 2,

‘ubiquinones are present in all organisms’

Because ubiquinone is not present in gram positive bacteria, this statement is not accurate. Refer ‘Kawamukai, Biosci. Biochem. Biotechnol. 2018’.

We removed the sentence containing this statement

Page 3,

Since the human genes for CoQ biosynthesis has been discovered as orthologues of the yeast genes, they should refer some works on yeasts such as ‘Hayashi et al PLoS one 2014’.

We added it as ref. 26 (Hayashi K, Ogiyama Y, Yokomi K, Nakagawa T, Kaino T, Kawamukai M. Functional conservation of coenzyme Q biosynthetic genes among yeasts, plants, and humans. PLoS One. 2014 Jun 9;9(6):e99038.)

Page 4,

‘no qui-nones were found bound to respiratory enzymes, leading to speculation of a dynamic equilibrium between quinone pools.’

 This statement should be fixed. The CoQ binding site in Complex I has been structurally solved in several species (Zickermann et al. Science 2015; Yoga et al., Front. Chem. 2021)

We referred the “bound” to a “Permament state of binding” (permanently bound, as modified in the text), meaning that there are not quinones which work as enzymatic prosthetic groups.

Pieces of Evidence  >> Pieces of evidence

Removed the part containing the typo

‘a D-lactate dehydrogenase quinone dependent is been recognized to be over-expressed’ >> a quinone dependent D-lactate dehydrogenase has been recognized to be over-expressed

Corrected

Page 5,

CytB5R3[70], For the reference of Cytb5R3, ‘Villalba et al. Mol. Aspects Med. 1997’ is better.

Changed

Page 7,

The reference numbers should be re-organized orderly through the text. Especially, the order of the references from 60 to 85 should be fixed.

Fixed all

‘In addition to the few examples given at the beginning of this chapter,’

Is this chapter ?

We removed this sentence

Page 8,

‘some authors’ > Quinzii et al.,

Changed “some authors” with the appropriate specification

Reviewer 3 Report

The authors reported the role of CoQ in maintaining the extra-mitochondrial electron transfer chain activity (Table 1) and the protective effect of CoQ in ferroptosis (Table 2) by self-citing their previous work. Furthermore, they tried to link their findings with developing targeted and effective therapies using the water-soluble form of CoQ (Phytosome formulation) and try to conquer the facing difficulty against extra CoQ supplementation for therapeutic purposes.

The reviewer thinks that this review manuscript is not well written. The followings are the reviewer’s opinions against the review manuscript.

(1) The authors did not describe why they thought to write this review and what they wanted to clarify about the current problems in the CoQ therapy or the finding the solid evidence of the clinical evidence of CoQ supplementation in the introduction section. Instead, they just wrote a list of sentences describing the history of CoQ. Unfortunately, the sentences did not remind the reading audiences what the authors insisted on in the main text.

(2) The reviewer did not understand why the authors showed their published data (Tables 1 and 2) in the main text. They are not necessary. In addition, these self-citations are not fair to write a review article. Any review articles should be introduced the recent findings widely and equitably. Even if they wanted to reinforce their findings in the main text, they should not include their original data; instead, they should illustrate the summary of their findings like Fig. 7 of reference 104. They should not explain their original data by halves. Only CoQ10 in the cell membranes section was described in detail, whereas other sections in the main text were not. The reading audience cannot understand why they described CoQ in the human diseases section. There is no story, just listing the sentences like the introduction section. The reading audience cannot catch the authors’ intention.

(3) The conclusion section was not led by the main text. The authors did not describe the low efficacy of CoQ10 supplementation for CoQ deficient patients, for example. The reviewer does not catch the meanings of the last sentence in the conclusion section, either.

(4) In the point of the abbreviation of Coenzyme Q, the authors used multiple abbreviations in the manuscript, CoQ, UQ, and COQ. They also used Coenzyme Q and coenzyme Q without using its abbreviation in the text repeatedly. It must make the reading audience confuse.

Author Response

Cover letter for the reviewer 2 of manuscript ijms-1461538

Dear Editor,

We would like to thank both reviewers for their helpful suggestions that lead us to modify (and hopefully improving) this present final version of the article.

Going through a point-by-point analysis, we modified the article following reviewers’ criticisms.

REVIEWER 2

The authors reported the role of CoQ in maintaining the extra-mitochondrial electron transfer chain activity (Table 1) and the protective effect of CoQ in ferroptosis (Table 2) by self-citing their previous work.

We cut out both tables

Furthermore, they tried to link their findings with developing targeted and effective therapies using the water-soluble form of CoQ (Phytosome formulation) and try to conquer the facing difficulty against extra CoQ supplementation for therapeutic purposes.

We added a new section on CoQ10 supplementation

  • The authors did not describe why they thought to write this review and what they wanted to clarify about the current problems in the CoQ therapy or the finding the solid evidence of the clinical evidence of CoQ supplementation in the introduction section. Instead, they just wrote a list of sentences describing the history of CoQ. Unfortunately, the sentences did not remind the reading audiences what the authors insisted on in the main text.

We re-organized the whole manuscript with the intention to drive the reader attention on the most recent issues regarding Coenzyme Q10 determination in biological samples, its use in therapeutic supplementation, and, briefly, on its role In disease.

  1. . Only CoQ10 in the cell membranes section was described in detail, whereas other sections in the main text were not. The reading audience cannot understand why they described CoQ in the human diseases section. There is no story, just listing the sentences like the introduction section. The reading audience cannot catch the authors’ intention.

We reorganized the manuscript with the intention of render it more balanced in the sections. Human disease section is organized on the bases of CoQ deficiency in a list of selected pathology, in order to focus on its role in therapeutic approach in diseases characterized by CoQ deficiency.

  1. The conclusion section was not led by the main text. The authors did not describe the low efficacy of CoQ10 supplementation for CoQ deficient patients, for example. The reviewer does not catch the meanings of the last sentence in the conclusion section, either.

Conclusion section has been reorganized and linked to the main text section. The last sentence of “Conclusion” section has been rephrased.

  1. In the point of the abbreviation of Coenzyme Q, the authors used multiple abbreviations in the manuscript, CoQ, UQ, and COQ. They also used Coenzyme Q and coenzyme Q without using its abbreviation in the text repeatedly. It must make the reading audience confuse.

We reduced the abbreviations of Coenzyme Q.

Round 2

Reviewer 1 Report

I suggest that the authors carefully check use of language in the added text sections as frequent problems with grammar are present

Author Response

We thank the reviewer for comments and for helping us to improve the manuscript.  The manuscript has  been reviewed and edited by a native English-speaking colleague. 

Reviewer 3 Report

Furthermore, they tried to link their findings with developing targeted and effective therapies using the water-soluble form of CoQ (Phytosome formulation) and try to conquer the facing difficulty against extra CoQ supplementation for therapeutic purposes.

We added a new section on CoQ10 supplementation (see main manuscript, page 9, line 452)

(1) The authors did not describe why they thought to write this review and what they wanted to clarify about the current problems in the CoQ therapy or the finding the solid evidence of the clinical evidence of CoQ supplementation in the introduction section. Instead, they just wrote a list of sentences describing the history of CoQ. Unfortunately, the sentences did not remind the reading audiences what the authors insisted on in the main text.

We re-organized the whole manuscript with the intention to drive the reader attention on the most recent issues regarding Coenzyme Q10 determination in biological samples (see main manuscript, page 7, line 354)  its use in therapeutic supplementation (see main manuscript, page 9, line 452) , and, briefly, on its role in disease. We stated in the abstract the purpose of the review (page 1, line 18): to report the latest hypotheses and theories analyzing the multiple functions of coenzyme Q, because the complete knowledge of the various cellular CoQ functions is essential to provide a rational basis for its possible therapeutic use.

The reviewer does not ask authors to describe them. This review article does not have the reason why they want to write this review article (theme of the review) in the introduction section and the conclusion what they want to inform to the reading audiences. They claimed that they stated their purpose in the lines 18-19 of the abstract. However, it is lacking in concreteness. Again, they did not state their purpose of the review article in the introduction section at all. The reviewer does not think that the authors responded to the comment.

(2) Only CoQ10 in the cell membranes section was described in detail, whereas other sections in the main text were not. The reading audience cannot understand why they described CoQ in the human diseases section. There is no story, just listing the sentences like the introduction section. The reading audience cannot catch the authors’ intention.

We reorganized the manuscript with the intention of render it more balanced in the sections. We shortened the Introduction section (Page 1, line 26), by moving some parts in the section “Ubiquinone in mitochondrial electron transfer chains”(Page 3, lines 115 to 161) and creating a new section entitled “Ubiquinone/Coenzyme Q biosynthesis”(page2, line 67). Human disease section ((see main manuscript, page 6, line 288) is organized on the bases of CoQ deficiency in a list of selected pathology, in order to focus on its role in therapeutic approach in diseases characterized by CoQ deficiency.

The theme of the review should be the relationship between CoQ functions and diseases. CoQ determination (lines ) is out of the theme of the review. Again, because their purpose of writing this review was not clarified in the introduction section, the reviewer thinks they just introduced the recent observation regarding CoQ cellular function probably related to human diseases. This manuscript is not an academic review article. It is like a general textbook. That is why the reviewer feels it does not have the authors’ viewpoint leading to their conclusion. The title of the review does not reflect the description in the main text.

(3) The conclusion section was not led by the main text. The authors did not describe the low efficacy of CoQ10 supplementation for CoQ deficient patients, for example. The reviewer does not catch the meanings of the last sentence in the conclusion section, either.

Conclusion section has been reorganized and linked to the main text section (see main manuscript, page 10, line 503). The last sentence of “Conclusion” section has been rephrased (see main manuscript, page 11, line 538).

See (2).

    In the point of the abbreviation of Coenzyme Q, the authors used multiple abbreviations in the manuscript, CoQ, UQ, and COQ. They also used Coenzyme Q and coenzyme Q without using its abbreviation in the text repeatedly. It must make the reading audience confuse.

We reduced the abbreviations of Coenzyme Q.

Coenzyme Q is still in the main text (Lines 79, 108, 156, 274, 356, 454, and 506). In addition, ubiquinone and ubiquinol should be unified to CoQ and reduced CoQ, respectively.

Author Response

We thank the reviewer for comments and for helping us to improve the manuscript. 

To better describe the intentions of our review, we have added a graphical abstract that more clearly illustrates our intentions. This review presents the issues related to coenzyme Q deficiency at the cellular level and its consequences, related to dysfunction of cellular homeostasis related to this deficit. Deficit that has not yet well-defined levels of quantification due to various technical problems in its determination and can not be completely resolved with administration of Coenzyme Q as a supplement because there are problems related both to the preparations used and the lack of complete knowledge of the mechanisms of distribution at the body level.